# Trehalose Activates Hepatic and Myocardial Autophagy and Has Anti-Inflammatory Effects in *db*/*db* Diabetic Mice

**DOI:** 10.3390/life12030442

**Published:** 2022-03-17

**Authors:** Tatiana A. Korolenko, Marina V. Ovsyukova, Nataliya P. Bgatova, Igor D. Ivanov, Svetlana I. Makarova, Valentin A. Vavilin, Alexey V. Popov, Ekaterina I. Yuzhik, Elena V. Koldysheva, Erik C. Korolenko, Evgeny L. Zavjalov, Tamara G. Amstislavskaya

**Affiliations:** 1Scientific-Research Institute of Neurosciences and Medicine, Timakov Street 4, 630117 Novosibirsk, Russia; maryov@ngs.ru (M.V.O.); amstislavskaya@physiol.ru (T.G.A.); 2Laboratory of Ultrastructural Research, Research Institute of Clinical and Experimental Lymphology—Branch of Institute of Cytology and Genetics, Siberian Branch of Russian Academy of Sciences, 630060 Novosibirsk, Russia; nbgatova@gmail.com; 3Laboratory of Drug Metabolism and Pharmacokinetics, Federal Research Center of Fundamental and Translational Medicine, Institute of Molecular Biology and Biophysics, Timakov Street 2, 630117 Novosibirsk, Russia; diadorych@mail.ru (I.D.I.); makar@niimbb.ru (S.I.M.); drugsmet@niimbb.ru (V.A.V.); popov_av@niimbb.ru (A.V.P.); 4Laboratory of Cytology and Cell Biology, Department of Molecular Mechanisms of Pathological Processes, Federal Research Center of Fundamental and Translational Medicine, Institute of Molecular Pathology and Pathomorphology, Timakov Street 2, 630117 Novosibirsk, Russia; pathol@inbox.ru (E.I.Y.); 130066@mail.ru (E.V.K.); 5Department of Quantitative Studies, West Pender Campus, University Canada West, 626 West Pender Street, Vancouver, BC V6B 1V9, Canada; erik.korolenko@ucanwest.ca; 6Institute of Cytology and Genetics, Siberian Branch of Russian Academy of Sciences, Academician Lavrentiev Avenue 10, 630090 Novosibirsk, Russia; zavjalov@bionet.nsc.ru

**Keywords:** leptin-deficient *db/db* mice, trehalose, autophagy, inflammatory dysregulation, *IL-10*, *TNF-alpha* expression, chitotriosidase, acid mammalian chitinase, lipophagy

## Abstract

*Db/db* mice (carrying a mutation in the gene encoding leptin receptor) show autophagy suppression. Our aim was to evaluate the effect of autophagy inducer trehalose on liver and heart autophagy in *db/db* mice and to study inflammation dysregulation and the suitability of chitinases’ expression levels as diabetes markers. Thirty-eight male *db/db* mice and C57/BL mice (control) were used. The *db/db* model manifested inflammation symptoms: overexpression of *TNF**-α* in the spleen and underexpression of *IL-10* in the liver and spleen (cytokine imbalance). Simultaneously, we revealed decreased expression of chitotriosidase (*CHIT1*) and acid mammalian chitinase (*CHIA*) in the liver of *db/db* mice. *CHIA* expression in *db/db* mice is significantly lower only in the spleen. Trehalose treatment significantly reduced blood glucose concentration and glycated hemoglobin. Treatment of *db/db* mice by trehalose was followed by increased autophagy induction in the heart and liver (increased autolysosomes volume density studied by morphometric electron-microscopic method). Trehalose exerted beneficial cardiac effects possibly via increased lipophagy (uptake of lipid droplets). The autophagy activation by trehalose had several positive effects on the heart and liver of *db/db* mice; therefore, lipophagy activation seems to be a promising therapy for diabetes.

## 1. Introduction

Type 2 diabetes is a prevalent metabolic disorder significantly contributing to morbidity and mortality in humans [1,2,3,4]. Several experimental models have been proposed for studies on Type 2 diabetes pathogenesis, among them *db/db* mice, which feature a leptin receptor deficiency [5]. New preventive measures and therapeutic agents have been suggested and developed for normalizing the glycemic profile in patients with Type 2 diabetes [6,7,8,9], including autophagy inducer trehalose, which is also effective in the *db/db* mouse model of diabetes [10]. The main mechanisms underlying the beneficial (hepatoprotective) action of trehalose were suggested recently and include: suppression of inflammatory signaling, enhancement of antioxidant defense, and induction of autophagy [11], although this topic needs further study.

It is believed that rare sugars (such as trehalose and others) hold promise for the prophylaxis and treatment of diabetes [12]. Some authors have demonstrated that rare sugars may serve as alternative sweeteners, especially for people who are at high cardiometabolic risk [13]. Nonetheless, these sugars have not been studied enough in experiments in vivo until now.

Trehalose, a disaccharide of glucose, is a naturally occurring nontoxic nonreducing bioactive sugar that is synthetized by many organisms when cells are exposed to stressful conditions, including dehydration, heat, oxidation, hypoxia, or anoxia [14]. Trehalose is not synthesized in the human body but is formed in many other organisms ranging from bacteria to plants; this compound may modulate insulin sensitivity via more than seven molecular pathways, thereby leading to better control of hyperglycemia [6,7,8,15]. These pathways are related to a number of major risk factors (although the exact causative mechanism has not been fully elucidated), including oxidative stress, inflammation, insulin receptor mutations, endoplasmic-reticulum stress, and mitochondrial dysfunction [16,17].

In an experiment on amyloid-β and islet amyloid pathologies, a link between Type 2 diabetes and Alzheimer’s disease was shown in a transgenic model [18]. Type 2 diabetes development has been linked with inflammation [19,20,21], although the underlying mechanisms are not fully understood currently [16]. Epidemiological studies have established an association between inflammatory biomarkers and complications of Type 2 diabetes. Adipose tissue appears to be a major site of production for these inflammatory biomarkers as a result of interactions among adipose cells, macrophages [22,23], and other immune cells that infiltrate the expanding adipose tissue in overweight patients and during atherosclerosis development [3]. In this regard, research on organs enriched with macrophages—the liver and spleen—is especially interesting in Type 2 diabetes [24,25]. In general, effects of trehalose in the heart have been poorly studied, although there are some data on a positive influence of trehalose in cardiometabolic diseases [26]. It was shown that trehalose upregulates cardiac autophagy marker LC3-II at 4 weeks after myocardial infarction, indicating that trehalose induces autophagy in the heart in vivo [27]. Further investigation is needed into the mechanisms behind trehalose effects in the heart in vivo.

Elmonem et al. [28] researched immune-inflammatory mechanisms underlying Type 2 diabetes development and revealed significant elevation of chitotriosidase (CHIT1) *activity*, a marker of activated macrophages. This enzyme was found to be secreted in patients with Type 2 diabetes in contrast to controls, while this parameter positively correlated with the progression of nephropathy (a frequent complication of diabetes). These authors were the first to demonstrate that the immunomodulatory effects of CHIT1 may be implicated in nephropathy development during Type 2 diabetes. Nevertheless, expression of chitinases has not been researched in patients with Type 2 diabetes; it is possible to investigate this issue in experimental models.

Later, another biomarker of Type 2 diabetes—YKL-40 (a protein without enzymatic activity, related to the chitinase family)—was found [17]. YKL-40 is an inflammatory glycoprotein participating in endothelial dysfunction by promoting chemotaxis, cell attachment and migration, reorganization, and tissue remodeling in response to endothelial damage. YKL-40 protein expression is detectable in macrophages and smooth muscle cells in diabetes and in atherosclerotic plaques with the highest expression seen in macrophages in Type 2 diabetes and in the early atherosclerotic lesion [29].

Chitotriosidase, belonging to the chitinase family, was suggested as a new marker of stimulated macrophages in diabetes [30]. Chitinases, chitotriosidase (CHIT1), acidic mammalian chitinase (AMCase or CHIA), and nonenzymatic chitinase-3-like protein 1 (CHI3L1) have been implicated in various pathological conditions, such as Gaucher’s disease (CHIT1), obesity, diabetes, cardiovascular diseases, and asthma (CHIA) [17,31]. The participation of these proteins in the pathological conditions has not been studied adequately until now. In general, as compared to CHIA, CHIT1 has been investigated more actively.

It is known that both *CHIT1* and *CHIA* are highly expressed in mice, while the level of *CHIA* mRNA is lower than that in human tissues [32]. According to a review by Lee et al. [33], mature monocyte-derived macrophages, Gaucher’s cells, and lung macrophages express *CHIT1*; proinflammatory cytokines (e.g., GM-CSF and TNF-α) and lipopolysaccharide (LPS) stimulate the expression of chitotriosidase in monocyte-derived macrophages, whereas IFN-γ and interleukin 4 (IL-4) inhibit *CHIT1* expression; CHIA attenuates chitin-induced innate inflammation, augments chitin-free allergen-induced T helper 2 (Th2)-based inflammation, and mediates effector functions of IL-13.

***The aim of this study*** was to evaluate the impact of autophagy inducer trehalose on liver and heart autophagy in *db/db* mice and to study inflammation dysregulation and the suitability of chitinases’ expression levels as diabetes markers.

## 2. Materials and Methods

### 2.1. Animals

C57BL/6 male mice (control) and *db/db* mice with leptin receptor deficiency (Pelletier et al., 2018) (aged 3 months at time 0 of acclimation) were purchased from the SPF-vivarium of the Institute of Cytology and Genetics SB RAS (Novosibirsk, Russia). The mice had access to standard mouse chow and water *ad libitum* during the 2-week acclimation period prior to experimentation and were maintained at 22 °C on a standard 12-h light/dark cycle. All procedures for the administration of compounds and blood and tissue collection were in accordance with the 8th edition of the Guide for the Care and Use of Laboratory Animals published in 2011 by the United States National Academy of Sciences, and the treatment protocol (animal protocol #9) was approved by the Institutional Animal Care and Use Committee of the Scientific-Research Institute of Neurosciences and Medicine. The in vivo experiments were also conducted in compliance with the Scientific-Research Institute of Neurosciences and Medicine Ethical Committee Recommendations pertaining to research involving laboratory animals.

### 2.2. Experimental Design

The mice were subdivided into four groups (6–8 animals each): (1) “C57BL/6 mice” (or “untreated C57BL/6 mice”), i.e., C57BL/6 mice drinking water *ad libitum* during the whole experiment (24 days); (2) “trehalose-treated C57BL/6 mice”, i.e., C57BL/6 mice drinking a 2% solution of trehalose (Trehalose dihydrate, Tokyo Chemical Industry, Japan) in water instead; (3) “*db/db* mice” (or “untreated *db/db* mice”), i.e., *db/db* mice drinking water *ad libitum* during the whole experiment; and (4) “trehalose-treated *db/db* mice”, i.e., *db/db* mice drinking the 2% trehalose aqueous solution instead of drinking water. On the day of euthanasia (day 25), the animals in each group were killed by decapitation. The scheme of the experiment is presented below (Figure 1).

### 2.3. Biochemical Assays

Murine blood was collected after decapitation, and serum was obtained by centrifugation on Eppendorf 5415R (Eppendorf AG, Hamburg, Germany) at 3000× *g* for 20 min. The serum was stored at −70 °C for ≤1 month and was used for enzyme assays and for the evaluation of the lipid profile. Fasting blood glucose, glycosylated hemoglobin HA1 [34], and liver function (ALT activity) were assayed during the experiment.

### 2.4. Differential Counting of Leukocytes

To determine differential counts of leukocytes, a drop of the blood was thinly spread on a glass slide, air dried, and subjected to Romanowsky staining by the May–Grunewald–Giemsa technique. Two hundred cells were then counted and classified.

### 2.5. Expression of IL-10, TNF-α, and Chitinases

The expression of chitinases [chitotriosidase (CHIT1) and acid mammalian chitinase (CHIA)] was assayed by a reverse-transcription quantitative polymerase chain reaction (RT-qPCR), as described earlier [35,36].

### 2.6. Isolation of RNA from the Liver and Spleen

Total RNA was isolated from a piece of tissue measuring 0.3 × 0.3 × 0.3 cm by guanidine thiocyanate–phenol–chloroform extraction with the LIRA reagent (Biolabmix, Novosibirsk, Russia), following the manufacturer’s protocol. The final dry RNA precipitate was diluted in 100 μL of RNAsecure and stored at −20 °C before use (for 2 months).

### 2.7. cDNA Synthesis

The amount of total RNA per reaction was 5 μg. Three microliters of random hexaprimers were added to the RNA template, and the volume of the mixture was brought to 12 μL with water. The mixture was heated at 70 °C for 2 min and then placed on ice. A mixture prepared beforehand was added, which contained 4 μL of OT-buffer-mix (Biolabmix, Novosibirsk, Russia) and 1 μL of M-MuLV-RH reverse transcriptase. It was mixed gently and incubated for 10 min at 25 °C and then for 60 min at 42 °C. The reaction mix was diluted to a total volume of 80 μL with distilled water, and then the reaction was stopped by heating 85 °C for 15 min and was held at −20 °C.

### 2.8. qPCR

A LightCycler 96 RT-PCR system (Roche, Basel, Switzerland) (thermal cycler) was employed to carry out qPCR. The reaction mixture was composed of 10 μL of BioMaster HS-qPCR SYBR Blue (2×) (Biolabmix, Novosibirsk, Russia), 400 nM primers, 3 μL of a cDNA template, and water up to 20 μL. The thermal cycling conditions were as follows: primary denaturation at 95 °C for 5 min, followed by 40 cycles of 95 °C for 10 s, 58 °C for 16 s, and 72 °C for 15 s. The primer sequences are presented in Table 1. *Eef2* and *Tbp* served as reference genes.

Relative mRNA levels were evaluated by means of C_t_ (threshold cycle) values by the 2^ΔCt^ method according to Livak and Schmittgen [37].

### 2.9. Morphological Investigations

#### 2.9.1. Light-Microscopic Examination of the Liver and Heart

All animals were killed by decapitation. After a necropsy and external examination, the liver and heart (which was placed in a cold chamber until complete cessation of the heartbeat) were weighed and cut with a sharp razor into pieces, one of which was fixed in a 10% solution of neutral formalin. The tissue samples for the preparation of semifine sections were fixed in a 4% paraformaldehyde solution with postfixation in a 1% solution of OsO_4_. The formalized samples were subjected to a standard procedure in an STP 120 histological tissue processor (Microm GmbH, Walldorf, Germany). Paraffin slices 2–3 mm thick obtained on an HM 325 microtome (Thermo Fisher Scientific, Runcorn, UK) were stained with hematoxylin and eosin as well as with Perls’ stain, van Gieson stain, and the periodic acid Schiff (PAS) reaction (OOO Ergoproduction, Saint Petersburg, LLC, Russia). The material, fixed in paraformaldehyde and intended for obtaining semifine slices, was processed according to the standard method and embedded into a mixture of Epon and araldite. Semifine slices 0.7–1 μm thick were stained with a 1% azure II solution. Paraffin and semifine sections were examined under a Leica DM 4000B universal research microscope (Germany). Micrographs were obtained by means of a Leica DFC 320 digital camera (Germany) and the Leica QWin3 software (Leica Microsystems, Cambridge, UK).

#### 2.9.2. Transmission Electron Microscopy

Myocardium and liver samples for electron microscopy were fixed in 4% paraformaldehyde in the Hanks medium and a 1% OsO_4_ solution (Sigma, St. Louis, MO, USA) in phosphate buffer (pH 7.4) for 1 h, dehydrated in ethanol of ascending concentrations, and embedded in Epon (Serva). Semifine 1 μm sections were prepared on a Leica EM UC7 microtome, stained with toluidine blue, and oriented for electron microscopy. Ultrafine sections with a thickness of 70–100 nm were contrasted with a saturated aqueous solution of uranyl acetate and lead citrate and analyzed under a JEM 1400 electron microscope (JEOL Ltd., Tokyo, Japan) (Multiple-Access Centre for Microscopy of Biological Subjects, Institute of Cytology and Genetics, Novosibirsk, Russia).

#### 2.9.3. Morphometric Electron-Microscopic Analysis

To determine volume density of autophagic structures in hepatocytes and cardiomyocytes, 30 cells in each group were randomly selected. Volume densities of autophagosomes, autolysosomes, and lysosomes were calculated using the ImageJ software (National Institutes of Health, Bethesda, MD, USA). Autophagic structures were identified according to the guidelines for monitoring autophagy [38].

#### 2.9.4. Morphometric Analysis of Lipid Inclusions

This analysis of cardiomyocytes was carried out on ultrafine longitudinal sections at ×12,000 magnification, the areas of lipid drop sections were measured, and their proportion in the total area of the sarcoplasm section (excluding nuclei) was evaluated; at least 10 nonoverlapping visual fields were analyzed for each animal. Quantitative data were obtained using the iTEM software (Olympus, Tokyo, Japan). Stereological analysis of the myocardium was performed as described earlier [39].

### 2.10. Statistical Analysis

Results are presented as the mean ± SEM and were subjected to one-way, factorial ANOVA followed by *post hoc* Fischer’s least significant difference test. Data with *p* < 0.05 were regarded as statistically significant. The normality of the data distribution was determined by the Kolmogorov–Smirnov test.

The fold changes of the relative mRNA levels between groups were determined by the 2^−ΔΔCt^ method, as described by Livak and Schmittgen [37].

## 3. Results

As reported before, compared to the control (C57BL/6 mice), the weight of *db/db* mice (drinking plain water) was greater because of obesity (*p* < 0.001, Figure 2), whereas trehalose consumption with drinking water decreased this parameter in comparison to untreated animals (*p* < 0.01, Figure 2). Heart weight was not different between untreated *db/db* mice and untreated C57BL/6 mice, whereas trehalose consumption decreased this parameter in *db/db* mice as compared to untreated mice (*p* < 0.05, Figure 3A). Liver weight was greater both in untreated (*p* < 0.001) and trehalose-treated *db/db* mice (*p* < 0.01) vs. respective controls (Figure 3B). The consumption of trehalose with drinking water decreased the body weight of *db/db* mice compared to untreated *db/db* mice, which is in agreement with our previous results [10]. Spleen weight was ~2-fold greater in untreated *db/db* mice than in untreated C57BL/6 mice (*p* < 0.001, Figure 4). This finding is possibly attributable to enhanced functioning of the spleen or even splenomegaly in Type 2 diabetes as a consequence of liver steatosis (Buchan et al., 2018).

In comparison with C57BL/6 mice, blood glucose concentration was significantly (*p* < 0.001) higher in *db/db* mice (Figure 5A), as was glycated hemoglobin, % (*p* < 0.001), (Figure 5B), whereas trehalose treatment significantly (*p* < 0.01) reduced both parameters (Figure 5A,B).

### 3.1. Blood Leukocyte Characteristics in Db/db Mice

#### The Impact of Trehalose Treatment on Peripheral-Blood Leukocytes in *Db/db* Mice

Trehalose consumption did not influence the numbers of polymorphonuclear leukocytes (PMNs) and lymphocytes in C57BL/6 mice, except for an increase in the monocyte number (*p* < 0.01, Figure 6). *Db/db* mice were found to have a greater relative number of PMNs (*p* < 0.001), a greater number of monocytes (*p* < 0.05), and a lower number of lymphocytes (*p* < 0.001) relative to C57BL/6 mice (Figure 6). Trehalose treatment of *db/db* mice decreased the PMN number (*p* < 0.01) and monocyte number (*p* < 0.01) while increasing the lymphocyte number (*p* < 0.05, Figure 6). 

### 3.2. Expression of TNF-α and IL-10 in the Liver and Spleen of Db/db Mice

In *db/db* mice, there was *TNF**-α* overexpression in the spleen (*p* < 0.001) and, to a lesser extent (*p* < 0.05), in the liver as compared to C57BL/6 mice (Figure 7A). We noticed *IL-10* underexpression in the liver (*p* < 0.001) and spleen (*p* < 0.01) of *db/db* mice (Figure 7B). This downregulation was more pronounced in the liver than in the spleen.

### 3.3. Expression of Chitinases in the Liver and Spleen of Db/db Mice

#### 3.3.1. Chitotriosidase and Acid AMCase in Diabetes

Chitotriosidase (CHIT1) and AMCase (CHIA) belong to the chitinase family. CHIT1 is the human chitinase studied the most regarding its biological activity and association with various disorders. In the healthy population, CHIT1 activity is very low and can originate from circulating PMNs [40,41].

#### 3.3.2. Expression of Chitinases in the Liver and Spleen of *Db/db* Mice

Compared to the control, *CHIT1* expression in the liver of *db/db* mice was significantly (*p* < 0.001) lower (Figure 8a), as was *CHIA* expression in the liver (*p* < 0.001, Figure 8c). In the spleen of *db/db* mice, there was *CHIA* underexpression (*p* < 0.01, Figure 8d) without aberrations in *CHIT1* expression in this organ (Figure 8b). Thus, both chitinases are underexpressed in the liver and spleen of *db/db* mice (with some specific features of *CHIT1* expression in spleen), which is in agreement with data on *different* expression in both chitinases (*CHIT1* and *CHIA*) shown in other experimental models [42].

### 3.4. Morphological Analysis of the Liver in Db/db Mice

#### 3.4.1. Light Microscopy

In C57BL/6 mice, liver structure overall was normal for small rodents, namely, hepatocytes formed radially arranged beam structures (Figure 9a). After the PAS reaction, diffusely located glycogen granules were detectable in stand-alone hepatocytes regardless of their localization and were sometimes present simultaneously with heterogeneous lipid inclusions (Figure 9e). The central veins were often found to be dilated and filled with blood. The administration of trehalose to the mice of the C57BL/6 strain did not cause significant changes in the architectonics of the liver (Figure 9b), but there was well-pronounced expansion of sinusoids and central veins. A slight increase in the amount of diffusely located glycogen (Figure 9f, arrows) was accompanied by a decrease in heterogeneous lipid inclusions (Figure 9f, asterisk).

In the liver of *db/db* mice, there was substantial structural and functional heterogeneity of the hepatocyte population: in the pericentral zone, hepatocytes were larger and possessed “empty” cytoplasm (Figure 9c); in the periportal zone, hepatocytes were smaller, and the cytoplasm stained with eosin uniformly. Most of the hepatocytes contained a considerable amount of lipid droplets, which were found to be depleted to varying degrees (underwent uneven lipolysis; Figure 9g, asterisk). All these characteristics developed against the background of microcirculatory disorders in the form of filling and pronounced dilation of large blood vessels in the triad system and of central veins. In the liver of trehalose-treated *db/db* mice, the hepatocytes looked more monomorphic; in their structure and tinctorial properties, these cells did not differ significantly between the pericentral and periportal zones (Figure 9d). In the assay based on the PAS reaction, it was obvious that glycogen occupied a large part of the hepatocytes’ volume (Figure 9h, arrows), while the number of lipid inclusions in most hepatocytes declined (Figure 9h, asterisk). In some cells, there were concurrent groups of lipid droplets and groups of glycogen granules (Figure 9h, arrows).

Thus, trehalose consumption for 24 days induced several significant structural changes in the liver: a noticeable decrease in lipid inclusions in the cytoplasm of hepatocytes, accompanied by an increase in the glycogen content. In addition, there was a decrease in the heterogeneity of the hepatocyte pool under the influence of trehalose intake.

In the liver of trehalose-treated *db/db* mice, the hepatocytes looked more monomorphic; in their structure and tinctorial properties, these cells did not differ significantly between the pericentral and periportal zones (Figure 9d). At the same time, it should be noted that there were persistent differences in the glycogen content between central and portal hepatocytes, albeit less pronounced than those in the mice that did not receive trehalose. In the assay based on the PAS reaction, it was obvious that glycogen occupied a large part of hepatocytes’ volume (Figure 9e), while the number of lipid inclusions in most hepatocytes declined markedly. In some cells, there were concurrent groups of lipid droplets and groups of glycogen granules (Figure 9f).

#### 3.4.2. Transmission Electron Microscopy

In *db/db* mice (compared with C57BL/6 mice, Figure 10), in the cytoplasm of hepatocytes, there was a higher content of large lipid inclusions (Figure 11a), a lower amount of glycogen, and a larger number of convoluted annular mitochondria (Figure 11b). In the cytoplasm of hepatocytes from trehalose-treated *db/db* mice, we observed autophagosomes with cytoplasm fragments, lipid inclusions, and autolysosomes with visualized lipid inclusions (Figure 11c–f).

#### 3.4.3. Morphometric Electron-Microscopic Analysis of Hepatocytes

The volume density of autolysosomes was threefold greater in hepatocytes in trehalose-treated *db/db* mice than in untreated *db/db* mice (Table 2, *p* < 0.05).

### 3.5. Morphological Analysis of the Heart

#### 3.5.1. Light Microscopy

Overall, the structure of the *myocardium* in the control mice was normal for small rodents. The nuclei of cardiomyocytes were situated centrally and were mostly large and euchromatic. In some cardiomyocytes, there was a moderate deficit of the sarcoplasm owing to lytic alterations. A distinctive feature of the cardiomyocyte structure in *db/db* mice was the presence of a large number of lipid droplets in their sarcoplasm (Figure 12c), which were overall evenly distributed within a cell.

Trehalose consumption did not induce any significant changes in myocardium structure, except for edema (Figure 12b). It is also important to note the altered nature of lipid infiltration of cardiomyocytes: it became mosaic instead of uniform, i.e., the number of lipid droplets in cardiomyocytes declined (Figure 12d), and hence a considerable area in each cardiomyocyte was free from lipid inclusions (Figure 12). These morphological data meant that autophagy was activated in order to mobilize lipid reserves for metabolic needs in the trehalose-treated group.

#### 3.5.2. Transmission Electron Microscopy

In the cardiomyocytes of *db/db* mice (drinking water) and after trehalose treatment, we noted accumulation of lipid inclusions, which were in close contact with mitochondria (Figure 12a,b). Autophagy of lipid droplets was registered infrequently in the control group, especially after the administration of trehalose (Figure 12c,d); there were autolysosomes and autophagosomes with mitochondria (Figure 12e,f).

#### 3.5.3. Morphometric Electron-Microscopic Analysis of the Heart

Volume density of autolysosomes in the cardiomyocytes was twofold greater in trehalose-treated *db/db* mice than in untreated *db/db* mice (Table 3, *p* < 0.05).

The lipophagy induced by trehalose in the heart may be a consequence of molecular interplay between autophagy and apoptosis (followed by a diminished rate of apoptosis) [43].

We can say that the lipid infiltration of cardiomyocytes was substantial. Electron-microscopic examination uncovered a compact packing of myofibrils and mitochondria in cardiomyocytes (Figure 13). The sarcoplasmic reticulum was represented by small vesicles, and the T-system mostly diminished. It should be pointed out that osmiophilic transformation of the content was seen in many lipid droplets: osmiophilic membrane structures emerged in them and were concentric.

Meanwhile, the cardiomyocytes of trehalose-treated mice were found to have a more distinct pattern of myofibrils and well-pronounced cristae of mitochondria, while the bulk density of lipids was 45% lower (Figure 14). In *db/db* mice, the morphological data after trehalose administration indicated greater efficiency of cardiomyocyte autophagy at ensuring the homeostasis of cardiomyocytes.

## 4. Discussion

The aim of this study was to evaluate the effect of autophagy inducer trehalose on liver and heart autophagy in *db/db* mice and to investigate inflammation dysregulation and the suitability of chitinases’ expression levels as diabetes markers. Earlier, we showed a positive effect of trehalose as an autophagy inducer in brain structures (hypothalamus and amygdala) with a significant improvement of performance in behavioral tests [10]. Nonetheless, it still was not clear what the effects of trehalose are in organs such as the liver and heart. It has been suggested that the effect of trehalose as an autophagy inducer may be more universal and occur in vivo in other organs. We demonstrated here that *db/db* mice, a model of diabetes, (aside from being overweight) are characterized by significant hyperglycemia and an increase of glycated hemoglobin, an elevated number of blood PMNs (a sign of inflammation) and monocytes, concurrently with a decreased lymphocyte number (immune dysregulation). These symptoms seen in the *db/db* model are generally typical for Type 2 diabetes in humans [44].

Type 2 diabetes development is connected with inflammation [45]. In our experiment with *db/db* mice, we obtained new data on increased pro-inflammatory *TNF-**α*
*expression* in the spleen and *decreased* expression of anti-inflammatory cytokine *IL*-*10* in the liver and spleen, as signs of a significant cytokine imbalance. Morphological analysis showed steatosis in the liver (lipidosis) and in the heart (greater formation of lipid droplets).

Trehalose treatment significantly reduced both blood glucose and the level of glycated hemoglobin. There was a liver ultrastructure improvement related to increased autolysosome volume density both in the liver and heart; we demonstrated this phenomenon for the first time by the electron microscopic morphometric method suggested by et al [38]. Using these methods, we detected an increase in autolysosome volume density and beneficial effects of trehalose in the liver and heart. The improvement in hepatocytes ultrastructure is possibly due to lipophagy after trehalose treatment. Increased autophagy in cardiomyocytes under the influence of trehalose treatment was demonstrated, too.

In our experiment, trehalose was used as 2% water solution. Approximately, per kg body weight of a mouse, the daily dose of dry trehalose was 3.2 g. When recalculated proportionately to human body weight of 70 kg, the daily dose of dry trehalose would be 224 g, which is fairly high compared to the recommended human daily intake of between 5 to 10 g.

### 4.1. The Db/db Mouse Model and Inflammation

Increased expression of proinflammatory cytokine *TNF*-α in the spleen and liver and decreased expression of anti-inflammatory cytokine *IL-10* in *db/db* mice, as a result of inflammatory dysregulation, were found in the present work. According to Akash et al. [46], among various proinflammatory cytokines, TNF-α is one of the most important and is crucial for the development of insulin resistance and for the pathogenesis of type 2 diabetes mellitus. TNF-α is mainly produced in adipocytes and/or peripheral tissues and induces tissue-specific inflammation by triggering the production of reactive oxygen species and the activation of various transcriptionally controlled pathways.

According to the results obtained in the present study, expression of *IL-10* is low in the liver and spleen of *db/db* mice, thus also confirming the signs of inflammation in this group; downregulation of *IL-10* was more pronounced in the liver than in the spleen (Figure 7A,B). IL-10 is considered an anti-inflammatory cytokine with lower circulating levels in patients with type 2 diabetes mellitus [47]. The main routine function of IL-10 appears to be to limit and ultimately terminate inflammatory responses. Such cytokines as IL-10 downregulate the production of proinflammatory cytokines, which impair the proper functioning of insulin [48].

Mediators of inflammation—TNF-α, IL-1β, the IL-6 family of cytokines, IL-18, and certain chemokines—are believed to be involved in the etiology of diabetes [45]. IL-6 is regarded as an important proinflammatory factor, and anti-IL-6 therapies have good clinical potential; their use may expand in the future [49]. In addition to immunoregulatory actions, IL-6 is thought to affect glucose homeostasis and metabolism directly and indirectly by acting on skeletal muscle cells, adipocytes, hepatocytes, pancreatic β-cells, and neuroendocrine cells [50]. IL-6 action is, in part, regulated by variants of IL-6 and IL-6α receptors and contributes to (but is probably neither necessary nor sufficient) for the development of Type 2 diabetes [51].

### 4.2. Expression of Chitinases in Db/db Mice

Chitinases are thought to be associated with inflammatory processes [52]. We showed for the first time that, in comparison with the control (C57BL/6 mice), *db/db* mice are characterized by decreased *CHIT1* expression in the liver (Figure 8a), while *CHIA* expression is lower both in the liver (Figure 8c) and spleen (Figure 8d). In C57BL/6 mice, relative expression levels of *CHIT1* and *CHIA* are similar between the liver and spleen (Figure 8). In *db/db* mice, *CHIA* was expressed more highly in the spleen (organ enriched with macrophages) than in the liver (enriched with macrophages—Kupffer cells).

Proteins from the family of chitinases and/or chitinase-like proteins play an important role in both innate and adaptive Th2 immune responses [33,53]. To date, more than seven members of this family have been identified in mice and humans, including enzymatically active chitinases: acid AMCase (AMCase, i.e., CHIA), chitotriosidase (CHIT1), and several chitinase-like proteins without enzymatic activity (e.g., oviductin, YKL-40/HcGP-39 [chitinase 3-like 1], and YKL-39) [33]. CHIA, CHIT1, and some other chitinases have been found in mice [54].

The chitinolytic enzymes analyzed in our work, namely CHIT1 and CHIA, have some similarities (in their characteristics) between humans and mice [31]. *CHIA* is expressed in epithelial cells and certain immune cells, such as neutrophils and macrophages in various organs of mice [55,56]. Human Kupffer cells are reported to oversecrete CHIT1 in steatohepatitis, and this phenomenon can be attributed to some complication of steatohepatitis [57,58,59]. By contrast, normal functions of CHIT1 have not been studied sufficiently until now [42].

Notably, it was demonstrated that CHIT1 can have a protective effect during the development of experimental atherosclerosis [60], whereas some authors reported increased serum CHIT1 activity in patients with atherosclerosis [61]. After administration of a CHIT1 inhibitor (allosamidin) in vivo, Artieda et al. [61] found that the inhibitor of CHIT1 (a protein synthesized exclusively by activated macrophages) exerted protective effects against atherosclerosis by suppressing inflammatory responses, polarizing macrophages toward the M2 phenotype, and promoting lipid uptake and cholesterol efflux in macrophages. Simultaneously, allosamidin inhibited the expression of scavenger receptor AI, CD36, ABCA1, and ABCG1, thereby leading to the suppression of cholesterol uptake and apolipoprotein AI-mediated cholesterol efflux in macrophages.

Until now, the other enzymatically active chitinase (CHIA) has been studied significantly less. Here, we report that *CHIA* expression in C57BL/6 mice is similar between the liver and spleen (Figure 8c,d), whereas in *db/db* mice, *CHIA* expression in the liver (Figure 8c) and *CHIT1* (Figure 8d) expression in the spleen are significantly lower than those in the control (C57BL/6 mice). According to a recent review [57], *CHIA* is expressed in epithelial cells, certain immune cells, neutrophils, macrophages, and in various organs (including the liver). Under physiological conditions, as a hydrolase, CHIA can degrade chitin-containing pathogens, participate in Th2-mediated inflammation, and enhance innate and adaptive immunity against an invading pathogen. Under some pathological conditions, *CHIA* expression needs further investigation. In general, CHIA has antiapoptotic activity, promotes epithelial-cell proliferation, and maintains organ integrity, and these actions are not related to the degradative function of the chitinase [57].

Animal models and clinical assessments of patients with diabetes have revealed that diabetes causes several anomalies at both biochemical and ultrastructural levels in several organs, including liver pathologies (steatosis and steatohepatitis) and heart damage.

Autophagy is an intracellular process designed to degrade dysfunctional proteins and damaged cellular organelles and regulates cell proliferation, differentiation, and apoptosis [1,62,63].

### 4.3. Expression of Chitinases in Db/db Mice, and Chitinases as Possible Markers of Diabetes

Chitinase (CHIT1 activity) was earlier suggested as a marker of Type 2 diabetes because increased CHIT1 *activity* in serum can serve as a biomarker of inflammation [30,64]. Nonetheless, the expression of chitinases has not been investigated enough in vivo, especially AMCase (CHIA) expression. In our work, we revealed underexpression of *CHIT1* and *AMCase* in the liver and spleen of *db/db* mice compared to C57BL/6 mice. Among two enzymatically active chitinases, AMCase (CHIA) is generally studied less than CHIT1. CHIA belongs to the 18-glycosidase family and is expressed in epithelial cells, certain immune cells, neutrophils, and macrophages in various organs [57].

Recently, increased *CHIT1* expression in adipose-tissue macrophages was shown in patients with Type 2 diabetes [21,22], pointing to the suitability of this metric as a potential biomarker of adipose-tissue inflammation. Nevertheless, there were no aberrations in *serum* CHIT1 enzymatic activity in the same patients with Type 2 diabetes. From the obtained results, it was concluded that *plasma* CHIT1 activity has only limited value as a circulating biomarker of adipose-tissue inflammation in humans.

Decreased expression of chitinases has also been demonstrated during progression of some tumors; for example, *AMCase* is aberrantly downregulated in gastric adenocarcinoma, liver cancer, renal clear cell carcinoma, and some other malignant tumors, but the specific mechanism is unclear [57,59] the physiological and pathophysiological relevance of chitinases in various organs opens up new possibilities for the treatment of diseases.

### 4.4. Possible Mechanisms behind the Protective Effects of Trehalose in Db/db Mice

Management of T2 diabetes includes the control of blood glucose and lipid levels and prevention of complications such as steatohepatitis, cardiovascular damage, and neurodegenerative alterations. Some compounds (β-glucans) have been proven to reduce risk factors of cardiovascular events, thereby improving the treatment of diabetes and of its complications [65]. Recently, trehalose, a known inducer of autophagy [10,66,67], was suggested as a possible therapeutic agent for the control of neurodegeneration and for the regulation of blood glucose levels. Trehalose treatment has significantly attenuated pathological changes in the hypothalamus (by inducing autophagy in this brain region) of mice and alleviated behavioral aberrations of mice in models of neurodegeneration. Studies of trehalose treatment indicate that autophagy stimulation is a potential strategy for overcoming Type 2 diabetic microvascular complications in *db/db* mice [68].

In our work, we showed a positive impact of trehalose consumption in *db/db* mice compared with untreated *db/db* mice. This impact was revealed as activation of lipophagy by trehalose in the mouse heart. According to recent data [69,70], autophagy activation in various ways can be useful for the prophylaxis and treatment of cardiovascular diseases, including atherosclerosis, coronary artery disease, Type 2 diabetic cardiomyopathy, arrhythmia, chemotherapy-induced cardiotoxicity, and heart failure. Trehalose, just as rapamycin, regulates the balance between cardiomyocyte apoptosis and autophagy in heart failure by inhibiting mTOR signaling [71].

### 4.5. The Therapeutic Effect of Trehalose on the Liver and Heart of Db/db Mice

This beneficial effect of trehalose treatment was successfully demonstrated in our study in liver morphological analyses and in a liver functional test: the ALT activity assay [10]. A similar effect was documented when exogenous trehalose administration inhibited atherosclerosis development and attenuated hepatic steatosis in apoE^−/−^ mice [72]. Several approaches have been developed to induce antisteatotic alterations in obese mice, and some of these modalities are: (i) the use of ACLY, an enzyme that regulates lipogenesis [73], and (ii) dapagliflozin and insulin glargine treatment [74,75].

Some authors explain the antisteatotic action of trehalose in the liver by an antioxidant effect of this compound [11] and by the induction of autophagy; however, further research is necessary to clarify this mechanism. A beneficial cytoprotective effect of trehalose treatment has been registered in numerous models of pathological processes, such as autophagy inhibition in the liver after Cd poisoning; for example, Cd-induced autophagy inhibition in liver cells was reversed by trehalose [76]. It was suggested in that report that trehalose treatment alleviated Cd-induced liver injury by blocking the NRF2 pathway, by restoring autophagy, and by inhibiting apoptosis.

The activation of autophagy by trehalose improves cardiac remodeling in heart diseases by increasing cardiac LC3-II levels [27]. Lipophagy (autophagic degradation of lipid droplets) is an important mechanism for preventing and managing the development of lipid storage syndrome in the liver and other tissues [77]. The prevention of T2 diabetes complications is important for the prophylaxis of cardiovascular diseases, liver damage, and kidney damage, which are associated with autophagy stimulation and lipophagy. Lipid droplets are key sites of neutral-lipid storage that can be found in all cells. Imbalances between the formation and degradation of lipid droplets can result in substantial lipid deposition: a characteristic feature of hepatocytes in patients with fatty liver disease [77]. Lipophagy has emerged as a major component of lipid metabolism with important implications for health [78].

Several rare sugars—allulose, arabinose, tagatose, trehalose, and isomaltulose—have been reviewed in the literature, and among them, trehalose [12], regarding benefits for glycemic control and weight loss in patients with diabetes and in healthy persons with obesity. These compounds hold promise for commercialization as alternative sweeteners, especially for individuals at high cardiometabolic risk. Trehalose, just as some rare carbohydrates (isomaltulose and d-tagatose) do, regulates glucose metabolism and supports glucose homeostasis in patients with diabetes but can also improve insulin sensitivity, subsequently leading to better control of hyperglycemia [13].

Trehalose can restore the functional autophagy suppressed by high glucose levels in vitro and in vivo [62]. Those authors showed that trehalose directly participates in the formation of functional autolysosomes by getting incorporated into these organelles. These findings provide the basis for applications of trehalose to the prevention of pathological autophagy-associated processes.

The mechanisms underlying the protective influence of trehalose include activation of hepatic transcription factor EB (TFEB); it was reported that trehalose protects against diet-induced nonalcoholic fatty liver disease in mice [79]. Namely, trehalose enhanced TFEB nuclear translocation and upregulated LC3-II and lysosomal proteins in the mouse liver (thereby confirming the activation of TFEB by trehalose).

## 5. Conclusions

The model of Type 2 diabetes in *db/db* mice is characterized by increased expression of TNF-α in the spleen, underexpression of IL-10 in the liver and spleen, an elevated number of PMNs in peripheral blood (a sign of inflammation), and diminished expression of chitinases in the liver and spleen. The downregulation of IL-10 in *db/db* mice reflects significant immunological disturbances in these animals, as demonstrated in humans with Type 2 diabetes. Trehalose treatment of *db/db* mice significantly reduced blood glucose concentration and glycated hemoglobin. Treatment of *db/db* mice by trehalose was followed by increased autophagy induction in the heart and liver (according to the increased autolysosomes’ volume density studied by the morphometric electron-microscopic method). We showed here that autophagy inducer trehalose in *db/db* mice has a positive effect on liver and heart morphology, possibly owing to the stimulation of autophagy in liver and heart cells. Lipophagy (related to autophagy) seems to play a considerable role in the overall protective mechanism in the heart and liver of *db/db* mice. Trehalose treatment, which induced lipophagy in heart and liver cells (and in brain cells, as was shown by us earlier), holds promise for the prevention and treatment of Type 2 diabetes in combination with other antidiabetic drugs. The prevention of Type 2 diabetes complications, including liver steatosis and cardiovascular diseases, is important for the prophylaxis of this disease. We suggest that in human trehalose can be used both in the prevention and treatment of cardiovascular diseases, Type 2 diabetes, liver steatosis, and some heart diseases.

## Figures and Tables

**Figure 1 life-12-00442-f001:**
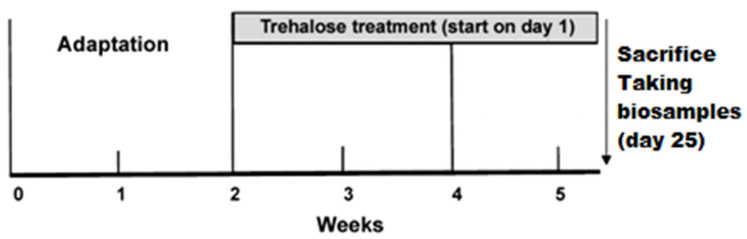
The scheme of the experiment.

**Figure 2 life-12-00442-f002:**
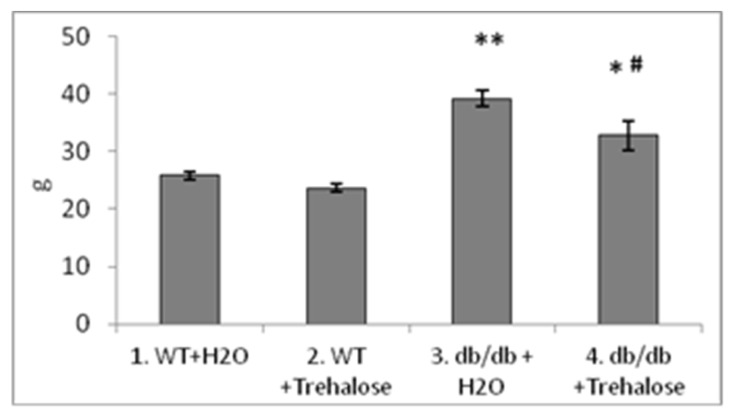
The influence of trehalose consumption with drinking water on body weight (g, mean ± SEM) of *db/db* mice. WT: C57BL/6 mice. ** *p*_1–3_ < 0.001, * *p*_2–4_ < 0.01, and ^#^
*p*_3–4_ < 0.05. The number of mice in each group is eight.

**Figure 3 life-12-00442-f003:**
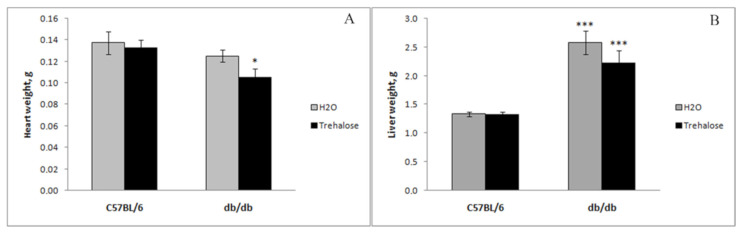
Effects of trehalose consumption on heart weight (**A**) and liver weight (**B**) of *db/db* mice (mean ± SEM). * *p* < 0.05 vs. untreated *db/db* mice (**A**), and *** *p* < 0.001 vs. a respective control, i.e., either untreated or trehalose-treated C57BL/6 mice (**B**). The number of mice in each group is eight.

**Figure 4 life-12-00442-f004:**
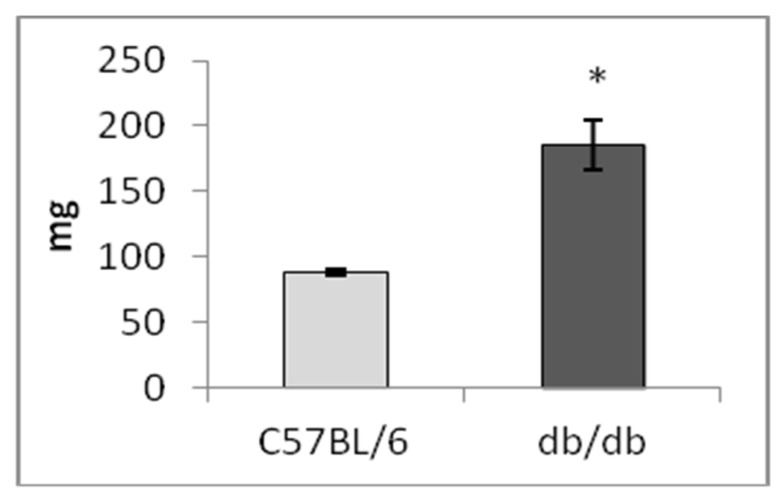
Spleen weight (mg, mean ± SEM) of *db/db* mice vs. the control (C57BL/6 mice). * *p* < 0.001 vs. control. The number of mice in each group is eight.

**Figure 5 life-12-00442-f005:**
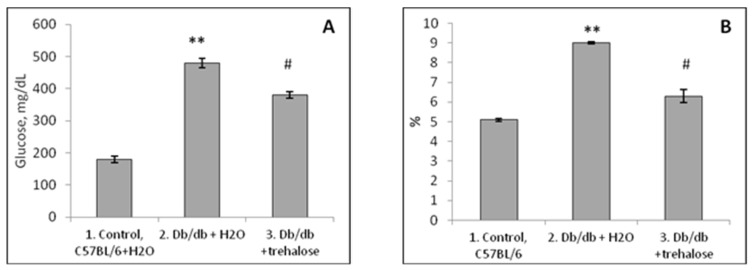
The influence of trehalose consumption on the blood glucose level (mg/dL) in *db/db* mice (**A**). ** *p* < 0.001 vs. C57BL/6 mice, and ^#^
*p* < 0.01 vs. untreated *db/db* mice. (**B**). Glycated hemoglobin (%, mean ± SEM) in the blood of *db/db* mice (**B**). ** *p* < 0.001 vs. C57BL/6 mice, and ^#^
*p* < 0.01 vs. untreated *db/db* mice. The number of mice in each group is eight.

**Figure 6 life-12-00442-f006:**
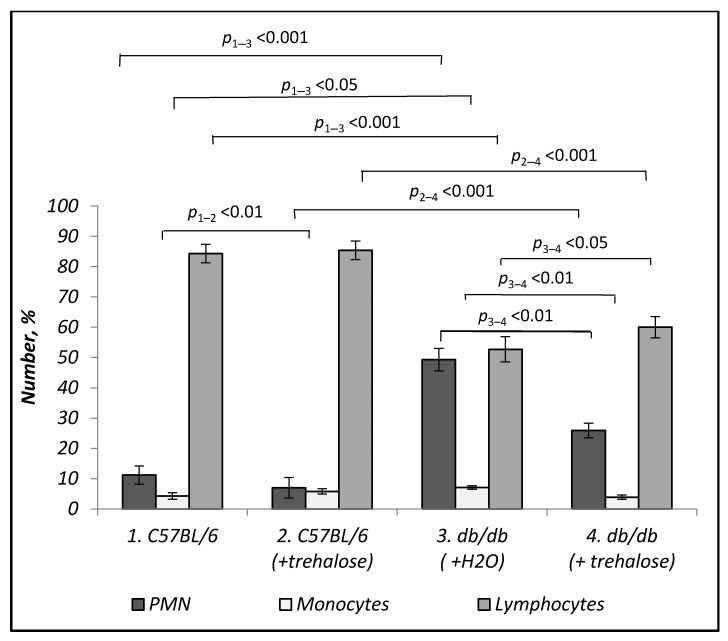
Effects of trehalose treatment on PMN, monocyte, and lymphocyte numbers in *db/db* mice. The number of mice in each group is eight.

**Figure 7 life-12-00442-f007:**
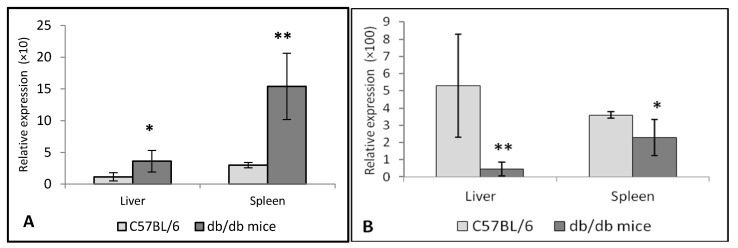
(**A**). Relative expression of the *TNF-α* gene in the liver and spleen of *db/db* mice. * *p* < 0.01; ** *p* < 0.001 vs. control (C57BL/6 mice). (**B**). Expression of *IL-10* in *db/db* mice. * *p* < 0.01 and ** *p* < 0.001 as compared with C57BL/6 mice. The number of mice in each group is eight.

**Figure 8 life-12-00442-f008:**
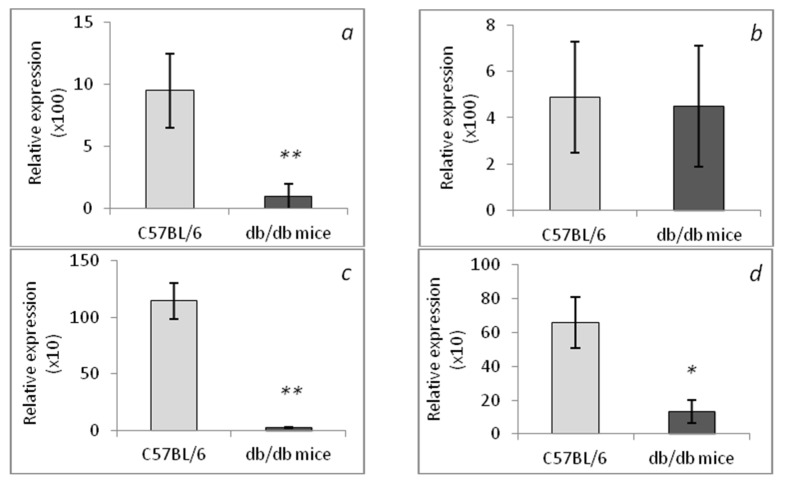
Relative expression of chitotriosidase (*CHIT1*) in the liver (**a**) and spleen (**b**) of *db/db* mice. Relative expression of *CHIA* in the liver (**c**) and spleen (**d**) of *db/db* mice. ** *p* < 0.001 and * *p* < 0.01 as compared with the control. The number of mice in each group is eight.

**Figure 9 life-12-00442-f009:**
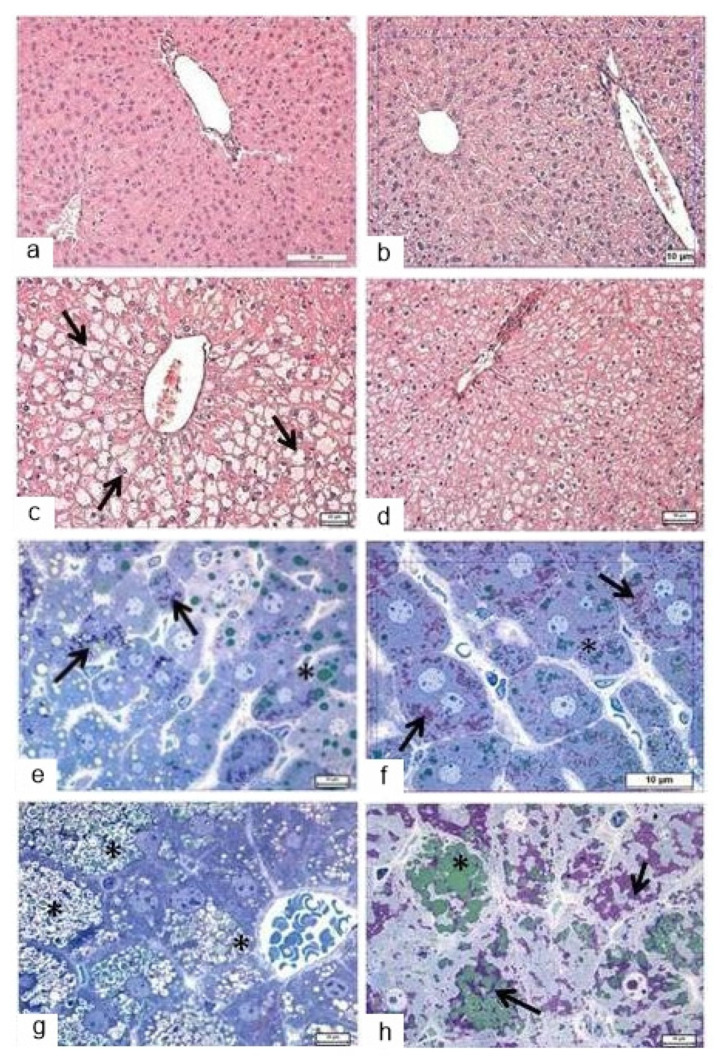
The influence of the consumption of 2% trehalose on liver histological characteristics. Liver morphology in C57BL/6 mice drinking either H_2_O (**a**,**e**) or H_2_O + trehalose (**b**,**f**), and in *db/db* mice drinking either H_2_O (**c**,**g**) or H_2_O + trehalose (**d**,**h**). Filling and pronounced dilation of large blood vessels in the triad system and of central veins (**a**,**b**). Large hepatocytes of the pericentral zone with “empty” cytoplasm (**c**, arrows), groups of glycogen granules (**e**, arrows), and lipid droplets with varying degrees of depletion (**e**, asterisk) in periportal hepatocytes. A slight increase in the amount of diffusely located glycogen (**f**, arrows) was accompanied by a decrease in lipid inclusions (**f**, asterisk). Glycogen occupies a large part of hepatocyte volume, and lipid inclusions are being depleted (**g**, asterisks). Groups of lipid droplets (**h**, asterisk) and groups of glycogen granules (**h**, arrows). Magnification: ×200 (**a**,**b**,**d**), ×400 (**c**), and ×1000 (**e**–**h**).

**Figure 10 life-12-00442-f010:**
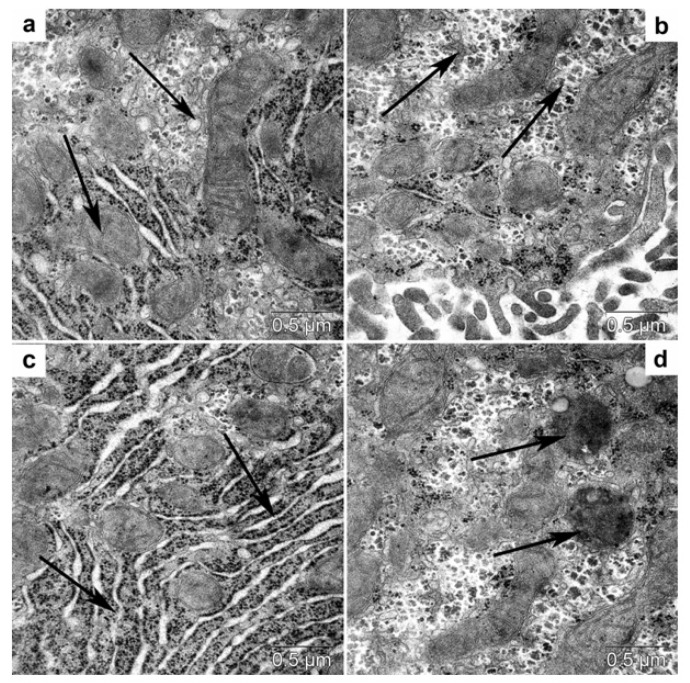
Ultrastructure of hepatocytes of C57BL/6 mice. (**a**) Mitochondria (arrows) with clear-cut cristae (**a**). Accumulation of glycogen (**b**, arrows). Cisternae of the granular endoplasmic reticulum (**c**, arrows). Lysosomes (arrows) in the hepatocyte cytoplasm (**d**).

**Figure 11 life-12-00442-f011:**
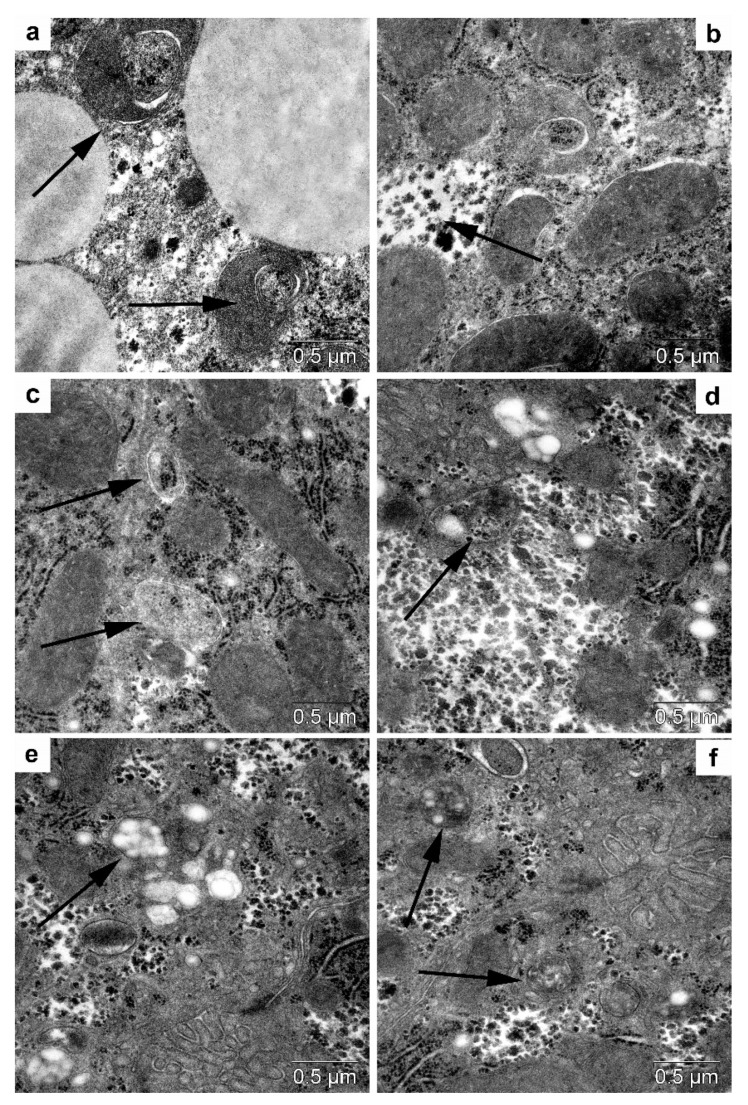
Hepatocyte ultrastructure in untreated *db/db* mice (**a**,**b**). Twisted mitochondria (arrows) and large lipid droplets (**a**), and a low glycogen content (**b**, arrows) in the cytoplasm of hepatocytes of *db/db* mice. Hepatocyte ultrastructure in trehalose-treated *db/db* mice (**c**–**f**). Autophagosomes with cytoplasm fragments (**c**,**d**, arrows), autophagosomes (**e**), and autolysosomes with lipid inclusions (**f**).

**Figure 12 life-12-00442-f012:**
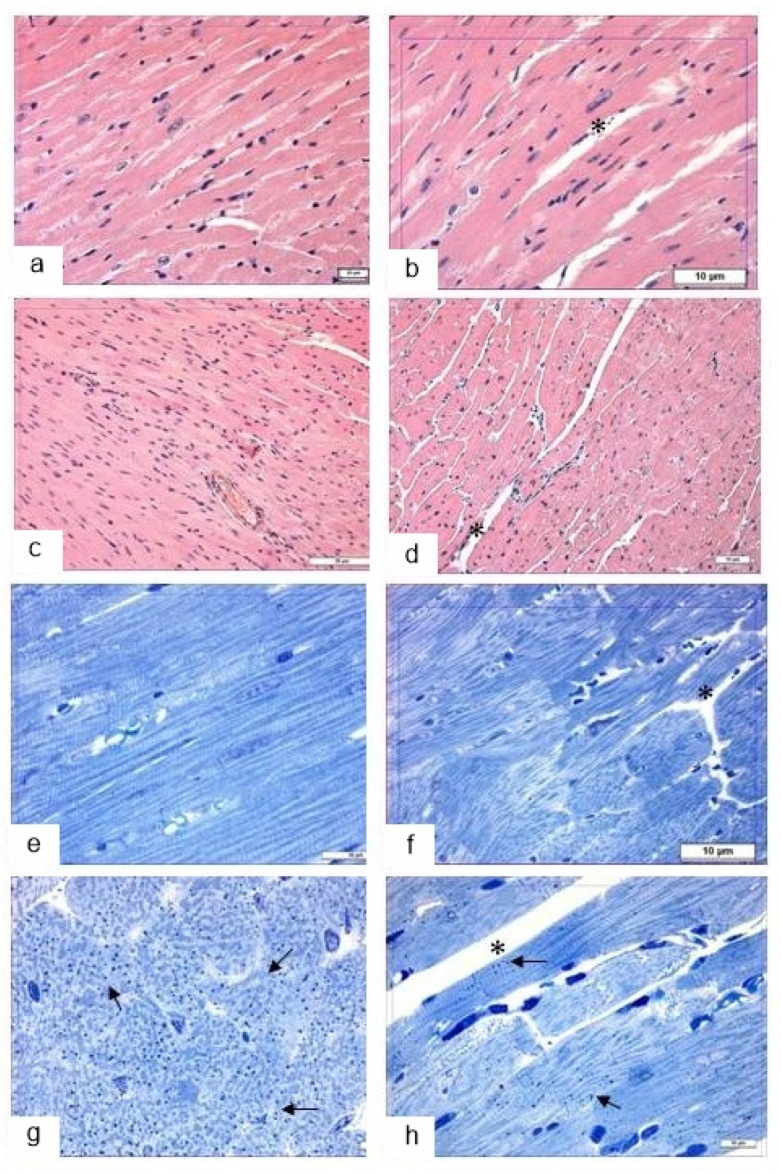
The influence of the consumption of 2% trehalose on heart morphology. Heart morphological analysis in C57BL/6 mice drinking either H_2_O (**a**,**e**) or H_2_O + trehalose (**b**,**f**) and in *db/db* mice drinking either H_2_O (**c**,**g**) or H_2_O + trehalose (**d**,**h**). The trehalose consumption resulted in pronounced edema (**b**,**d**,**f**,**h**; asterisk). Lipids in cardiomyocytes (arrows) in *db/db* mice (**g**) and trehalose-treated *db/db* mice (**h**). Magnification: ×200 and ×1000. Magnification: ×200 (**c**,**d**), ×400 (**a**,**b**), and ×1000 (**e**–**h**).

**Figure 13 life-12-00442-f013:**
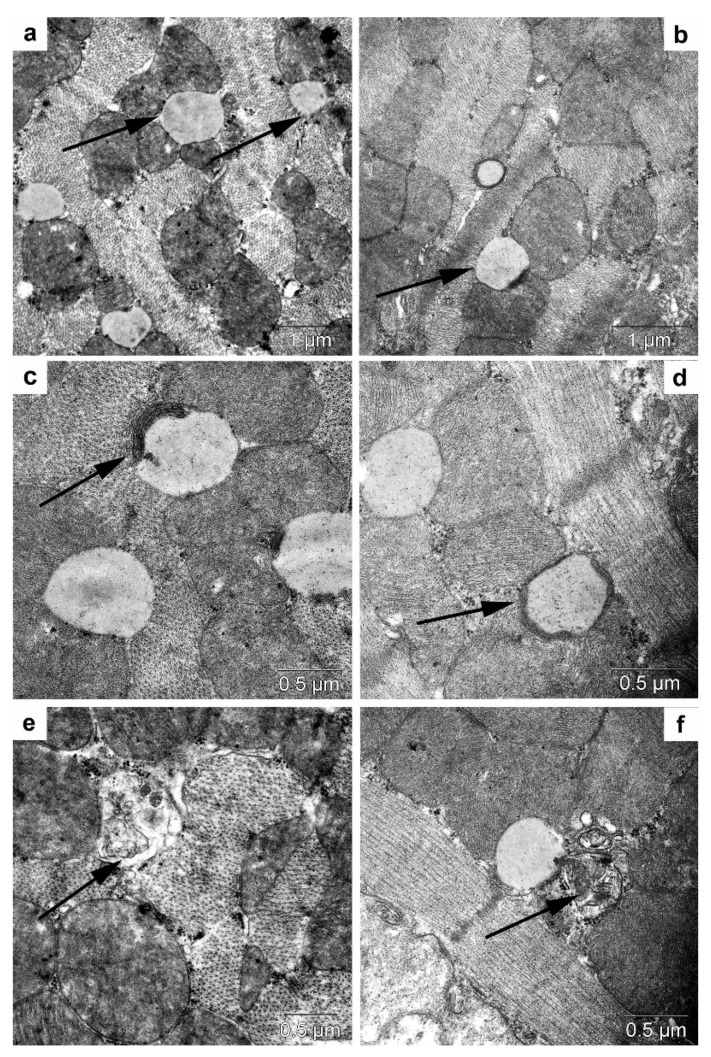
The ultrastructure of cardiomyocytes in *db/db* mice. Lipids in a cardiomyocyte (arrows) in the control group (**a**) and after trehalose administration (**b**). Separation of lipids into an autophagosome (arrows) in the control (**c**) and after trehalose administration (**d**). An autolysosome (arrow) in a cardiomyocyte in the control group (**e**) and an autophagosome with mitochondria (arrow) after trehalose administration (**f**).

**Figure 14 life-12-00442-f014:**
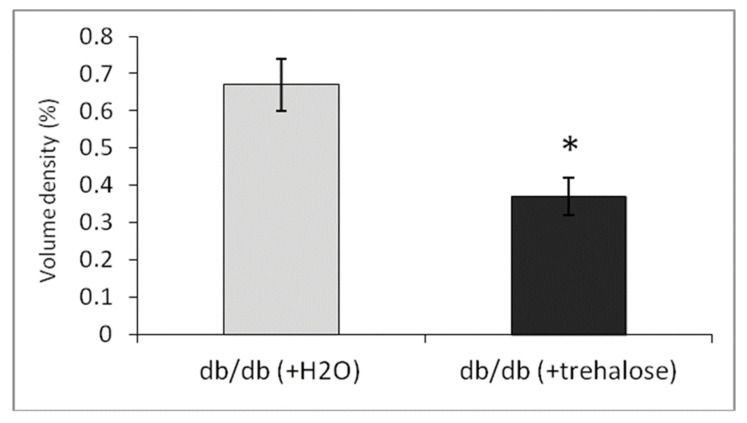
The impact of trehalose on volume density (mean ± SEM) of lipid inclusions (%) in cardiomyocytes of *db/db* mice. * *p* < 0.05 as compared with untreated *db/db* mice.

**Table 1 life-12-00442-t001:** Primers for qPCR.

Target	Oligonucleotide Sequence
*Mus musculus EEF2*	forward 5′-GGAGACCGGTACTTTGATCC-3′
	reverse 5′-TTTGTCCTCGCTGTCCAGCT-3′
*Mus musculus TBP*	forward 5′-AGGAGCCAAGAGTGAAGAACAA-3′
	reverse 5′-ATAATTCTGGCTCATAGCTACTGA-3′
*Mus musculus CHIA*	forward 5′-TCCTGGTGAAGGAAATGCGT-3′
	reverse 5′-AAATCCCACCAGCTACAGCA-3′
*Mus musculus CHIT1*	forward 5′-TCAGACAATGGAGTTGGGGC-3′
	reverse 5′-TTCCAGGAGCAGGCCTCATA-3′
*Mus musculus IL-10*	forward 5′-TGGGTTGCCAAGCCTTATCG-3′
	reverse 5′-CTCTTCACCTGCTCCACTGC-3′
*Mus musculus TNF-*α	forward 5′-TCAGTTCTATGGCCCAGACC-3′
	reverse 5′-ACCACTAGTTGGTTGTCTTTGAG-3′

**Table 2 life-12-00442-t002:** Volume density (mean ± SEM) of autophagic structures in hepatocytes of *db/db* mice.

Groups of Mice	Autophagosome, V_v_	Autolysosome, V_v_	Lysosome, V_v_
*db/db* + H_2_O	0.67 ± 0.18	0.4 ± 0.21	1.31 ± 0.97
*db/db* + trehalose	0.91 ± 0.32	1.26 ± 0.99 *	1.56 ± 0.26

* A significant difference of trehalose-treated *db/db* mice from untreated *db/db* mice (*p* < 0.05). V_v_: volume density of structures (%).

**Table 3 life-12-00442-t003:** Volume density (mean ± SEM) of autophagic structures in cardiomyocytes of *db/db* mice.

Groups of Mice	Autophagosomes, V_v_	Autolysosomes, V_v_	Lysosomes, V_v_
*db/db* + H_2_O	2.58 ± 0.41	0.99 ± 0.32	0.01 ± 0.008
*db/db* + trehalose	2.28 ± 0.30	2.07 ± 0.34 *	0.02 ± 0.009

* A significant difference in the *db/db* + trehalose group from the untreated *db/db* group, *p* < 0.05. V_v_: volume density of structures (%).

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
