# Peer review of "Trehalose Activates Hepatic and Myocardial Autophagy and Has Anti-Inflammatory Effects in db/db Diabetic Mice"

_life, 2022, doi:10.3390/life12030442_

Round 1

Reviewer 1 Report

The authors have done experiments to answer the previous questions. In general, I think the discovery is interesting and most of the functional studies are convincing. I only have a few points for the authors.

  1. Figure 2 and figure 5 are not complete
  2. Please carefully check all the description. There still are some errors in this vesion. For example, LINE 200, 253, 268, 269, 314, 459, 474, and 478.

Reviewer 2 Report

The authors describe a thorough examination of the effects of trehalose on outcome measures important to diabetes. The writing is clear and the figures and tables are effective, although some minor revisions are suggested.

Materials, Methods:

Indicate the age of the mice at time 0 of acclimation.

What is the approximate dosage of trehalose on a kg body weight basis and how does this equate to human consumption?

Figures 2 needs X and Y axis labels and tick marks, and group labels. Figure 5 is also incomplete.

Use the same layout, size and unit of measure (g) for all organ figures (3, 4).

Figure 6: Change "The number" to "Number" or "percentage of ___"

Figure 9: Suggest using letter identifiers of the images that are similar to those in figures 10 & 11. Currently, Figure 9 letters are difficult to see. Use upper- or lower-case letters consistently.

Include the number of animals in the footnote of each figure.

Discussion:

Line 259-60 sentence structure needs to be corrected.

Please add a discussion item of how the dose and duration of trehalose in the animals translates to human consumption. Also include, if the information is available, the current consumption levels in human populations.

This manuscript is a resubmission of an earlier submission. The following is a list of the peer review reports and author responses from that submission.

Round 1

Reviewer 1 Report

The authors describe a number of cellular and histological changes in the diabetes model mice db/db upon feeding them with a disaccharide trehalose. Changes in the expression levels of TNF, IL-10, CHIT1 and CHIA1 were observed. Some changes in cellular and tissue morphology took place in livers and hearts of the treated animals.

There are major scientific issues regarding the scientific and formal aspects of the manuscript.

  1. There are no specific data (autophagy markers) provided that trehalose treatment as applied induced or enhanced autophagy in the analyzed tissues.
  2. The analyses of liver and heart cells are performed in db/db mice only (Figure 5-8). The comparison with the respective tissues from control mice is lacking.
  3. There is no quantification of morphological differences provided (besides the volume density of lipid inclusions in Figure 9). At least semi-quantitative parameters should be evaluated and compared to this end. The authors indicate some possibilities already, e.g., “lipid droplets were found not in all cardiomycytes but rather in approximately half of them […]” (Section 3.5.1).

The language of the manuscript and the presentation of the data are below the usual standard. Even the title and abstract contain spelling, punctuation and semantic mistakes. This makes reading the manuscript a strenuous undertaking. Furthermore, the authors are referring extensively to their previous data or data of others in the Results section (e.g., the last paragraph of 3.1.1., the entire 3.2.1., the second half of 3.2.2., the entire 3.3. etc.). This information belongs either to the Introduction or the Discussion. A significant editing of the text is necessary to improve the presentation of the study.

Reviewer 2 Report

The authors tried to clarify an interesting question by which tregalose activates hepatic and myocardial autophagy and demonstrates anti-inflammatory effects in db/db diabetic mice. In this study, they observed some characters in liver and heart tissue from the db/db mice after tregalose treatment. However, this study over-interpreted the results and lacked conclusive evidence to support their hypothesis. The whole article needs to be reorganized, and too much unnecessary information is described in this manuscript (like the paragraph of neurodegenerative diseases). Furthermore, the levels of autophagy should be quantified in all imaging studies. There is no evidence by which tregalose regulates inflammation response in this study.

Reviewer 3 Report

The authors contribute important data to the field of treatment/prevention of diabetes and its co-morbidities. The research methods appear appropriate for testing the effects of trehalose on outcomes.

Comments:

Extensive English editing is required. Errors in sentence structure, use of punctuation and use of articles (a, the) are seen throughout the manuscript. Semi-colons, especially, are used inappropriately.

Generally, the results section should include only outcomes data and not commentary. Move your discussions of how your data compare to other studies' to the Discussion section.

Citation and bibliography formats need to adhere to the journal's policies.

Avoid the used of "thanks" and replace with "due to".

Some statements are missing supporting references:

Sentence starting: "Earlier we have shown that treatment by autophagy..."

"Trehalose, a disaccharide of glucose, is a naturally occurring nontoxic and nonreduc-ing bioactive sugar that is synthetized by many organisms when cells are exposed to stressful conditions, including dehydration, heat, oxidation, hypoxia..."

Results: "As reported before..."

"...development of hepato-spleen syndrome..."

"The functions of autophagy in ensuring cellular survival and in suppression of neurodegeneration have been evaluated in Alz-heimer’s, Parkinson’s, and Huntington’s diseases, which are accompanied by the accu-mulation of beta-amyloid, alpha-synuclein, and huntingtin, respectively."

"Under some pathological conditions (type 2 diabetes mellitus, chronic atrophic gastritis, and pulmonary interstitial fibrosis), CHIA expression significantly changes."

"Animal models and clinical assessments of patients with diabetes have revealed that diabetes causes several abnormalities at both the chemical and the ultrastructural levels of the brain, leads to dysfunctional behaviors, and raises the risk of major depressive dis-order. Although accumulating evidence indicates that oxidative stress, chronic inflammation, and programmed cell death may cause the brain dysfunction associated with many neurological diseases, the exact mechanisms underlying diabetes-related neurological anomalies remain unclear."

Materials and methods:

Describe the age of the mice. Were the groups housed together or were mice housed individually? How often was body weight measured? Were intakes of food and water measured? Water intake measures are potentially important in considering outcomes.

State the source of trehalose.  How often was water prepared and fresh water replenished?

Describe the chow diet and provide the composition in supplementary materials.

Biochemical assays: describe the assays and provide information on the methodology.

Figure 1. Y axis label should state Spleen weight (mg)  and specify if wet weight. Provide number of animals in each group in the footnote for this and all figures/slides.

Figure 2. Y axis label should state Relative number (%). Specify in the text and figure footnote how this was calculated (relative to what?) Correct the X axis labels tregalose vs. trehalose.

Report the body weight data.

Show data associated with these statements: "As shown earlier (Korolenko et al., 2021), blood glucose concentration proved to be significantly elevated in db/db mice vs. control C57BL/6 mice, and trehalose treatment de-creased the blood glucose level compared to untreated db/db mice, albeit not to the appropriate level seen in the control C57BL/6 mice."

3.2.1. Chitinases and Diabetes - these statements belong in the introduction and not results.

Figure 3 footnote: Explain how the relative value was calculated. Specify "...of db/db and C57BL/6 mice..." for both parameters.

Figure 4: create panels A and B. Make the figure design/format the same for both panels. Specify "...of db/db and C57BL/6 mice..." for both parameters.

The paragraphs beginning "IL-10",  "IL-6" and "According to Akash et al." are not results and belong in the introduction or discussion.

Figure 5. State the meaning of the arrows and add a scale bar as done in figure 6.

Figure 7: add scale bar.

Discussion:

Begin with a summary of your findings. Then proceed to speculate on the significance and causal factors related to these.